# INFORM-CT: INtegrating LLMs and VLMs FOR Incidental Findings Management in Abdominal CT

**Idan Tankel**[*1]

[1] *GE Healthcare Technology and Innovation Center, Niskayuna, USA*

**Nir Mazor**[*1]

**Rafi Brada**[1]

**Christina Lebedis**[2] (iD)

[2] *Boston Medical Center, Boston, USA*

**Guy Ben-Yosef**[†‡1] (iD)

**Editors:** Accepted for publication at MIDL 2026

## Abstract

Incidental findings in CT scans, though often benign, can have significant clinical implications and should be reported according to established guidelines. Traditional manual inspection by radiologists is time-consuming and subject to variability.

This paper proposes a novel framework that leverages large language models (LLMs) and foundational vision–language models (VLMs) within a plan-and-execute agentic architecture to improve the efficiency and precision of incidental-findings detection, classification, and reporting in abdominal CT scans. Given medical guidelines for abdominal organs, the management process is automated through a planner–executor framework. The planner, based on an LLM, generates Python scripts from predefined base functions, while the executor runs these scripts to perform the required detections and evaluations using VLMs, segmentation models, and image-processing subroutines.

We demonstrate the effectiveness of our approach through experiments on a CT-abdominal benchmark covering three organs, in a fully automatic end-to-end setup. Our results show that the proposed framework outperforms existing purely VLM-based approaches in both accuracy and efficiency. Implementation details and code are available at this repository.

**Keywords:** Incidental Findings Detection, Abdominal CT, Vision-Language Models, Planner-Executor Framework, Clinical Guidelines

## 1. Introduction

Incidental findings on abdominal CT scans are common and may have important clinical implications. Therefore, it is crucial to report these findings in an actionable manner, adhering to established guidelines. Virtually every scan reveals incidental findings, making it essential to distinguish significant findings from background noise. This paper aims to address the clinical concern of managing the overwhelming number of findings, especially in older individuals where incidental findings are prevalent. We propose a novel framework based

---

[*] Contributed equally

[†] Contributed equally

[‡] Corresponding Author

on LLM combined with VLM in an agent framework, particularly of a plan-and-execute style, to improve the efficiency and precision of automatic incidental findings analysis in abdominal CT imaging, adhered to medical guidelines.

Traditional methods for incidental findings detection on abdominal CT rely on manual inspection by radiologists, which can be time-consuming and prone to variability (Berland et al., 2010). In the past decade, deep learning-based medical anomaly detection has emerged as a relevant approach. These methods often aim to learn the distribution of normal patterns from healthy subjects and detect anomalous ones as outliers, for instance, via autoencoders or generative adversarial networks (e.g., (Zhang et al., 2023; Schlegl et al., 2019; Akcay et al., 2019; Shvetsova et al., 2021; Almeida et al., 2023)). Other relevant models target segmentation or detection of specific types of incidental findings (e.g. liver mass (Lyu et al., 2024)). However, none of these methods propose a general-purpose approach for detecting multiple incidental findings across various organs, such as in abdominal imaging.

Vision-language multimodal approaches have shown promise in enhancing the detection of pathologies by leveraging both visual and textual information. CLIP (Radford et al., 2021) efficiently learns visual concepts from natural language supervision, enabling zero-shot transfer capabilities. For medical 3D inputs, CT-CLIP (Hamamci et al., 2024) and BIMVC (Chen et al., 2024) focus on chest CT volumes, pairing them with radiology text reports to improve diagnostic accuracy. MERLIN (Blankemeier et al., 2024), designed for abdominal CT, integrates textual and 3D visual data to provide comprehensive insight into abdominal imaging. These models collectively advance the field of medical imaging by combining visual and textual information, improving zero-shot classification tasks without additional annotations. However, these models still struggle to perform complex diagnostic tasks, and as we show here, they can be significantly augmented when paired with LLMs and computer vision sub-routines in an agent-based framework.

Planner-executor systems automate complex tasks by generating and executing code based on pre-defined instructions. Recent advances in plan-and-execute frameworks have paved the way for the integration of LLM-powered agents. These agents can plan and perform actions, enhancing the overall efficiency and accuracy of task execution. The majority of computer vision work for such systems focuses on visual question answering (VQA). For example, models such as (Surís et al., 2023; Khan et al., 2024; Gupta and Kembhavi, 2023) leverage code-generation models as well as vision-language models such as CLIP into subroutines, producing results for any query by generating and executing Python code. More advanced methods integrate a planner, reinforcement learning agent, and reasoner for reliable reason (e.g., (Ke et al., 2024)) or use a multi-turn conversation and feedback (e.g., (Yao et al., 2023; Min et al., 2024)). In the context of incidental findings detection, such systems can ensure the adherence to clinical protocols and improve the efficiency of the inspection process. To our knowledge, we are the first to apply this approach of code generation and execution for CT diagnosis, providing a novel and interpretable solution for medical imaging analysis.

While prior VQA-based approaches typically rely on a small set of base functions (e.g., object detectors, CLIP) and produce short programs, our setting requires substantially more complex programs together with low-level image processing primitives (e.g., size, edge,

intensity), which motivates a careful design of the underlying plan-and-executor architecture tailored to the medical imaging domain.

To conclude, our contributions in this paper are as follows:

(i) We are the first to propose an incidental findings pipeline for the entire abdominal region, based on an LLM and VLM agentic approach. This pipeline is general, automatically created, and adheres to clinical protocols and guidelines.

(ii) We propose a *plan-and-execute* program generation method, which starts from a `PDF`, and automatically generates and executes a robust Python program with multiple visual subroutines (base functions) that predict clinical recommendations.

(iii) We introduce a benchmark and a new method to create test examples for incidental-finding recommendations, based on Abdominal-CT reports.

## 2. Method

The proposed method aims to automate the management of incidental findings on abdominal CT scans for multiple abdominal organs, based on `PDF`s of medical guidelines. This entire process is performed end-to-end automatically using our planner-executor framework. The framework utilizes the parsed guidelines (stored in a `JSON` file) and available protocols to generate and execute the necessary code for inspection. An overview of the full pipeline is shown in Figure 1.

### 2.1. Parsing Guidelines

We begin by parsing the medical guidelines, which often come in `PDF` format, into decision trees that include multiple checks and detections leading to recommendations. For this parsing stage, we used LLM (GPT-4o (OpenAI, 2023)), and the LangChain framework (Chase, 2022), to analyze figures, tables, cross-references, footnotes, and `PDF` text, converting them into `JSON` formats applicable for later stages. An example of a `PDF` and the parsed tree is shown in Figure 2. The parsed `JSON` file contains structured information extracted from the guidelines, including checks, detections, measurements, and recommendations. This structured format allows for easy integration into the planner-executor framework.

### 2.2. Planner-Executor Framework

Using the parsed guidelines (stored in a `JSON` file) and available protocols, we implemented a planner-executor framework:

- *Planner*: The planner, a ReAct (Yao et al., 2023) agent set up on Claude 3.5 (Anthropic, 2024) (selected for its strong code generation capabilities), generates a Python script using a set of predefined base functions. It utilizes the parsed guidelines to create the script.

- *Executor*: The executor runs the generated Python script, triggering the inner base functions.

At the core of this framework is the code generation of a Python script designed to inspect incidental findings based on medical guidelines. The challenge lies in the complex structure of the decision trees from Section 2.1 and the variety of visual subroutines involved

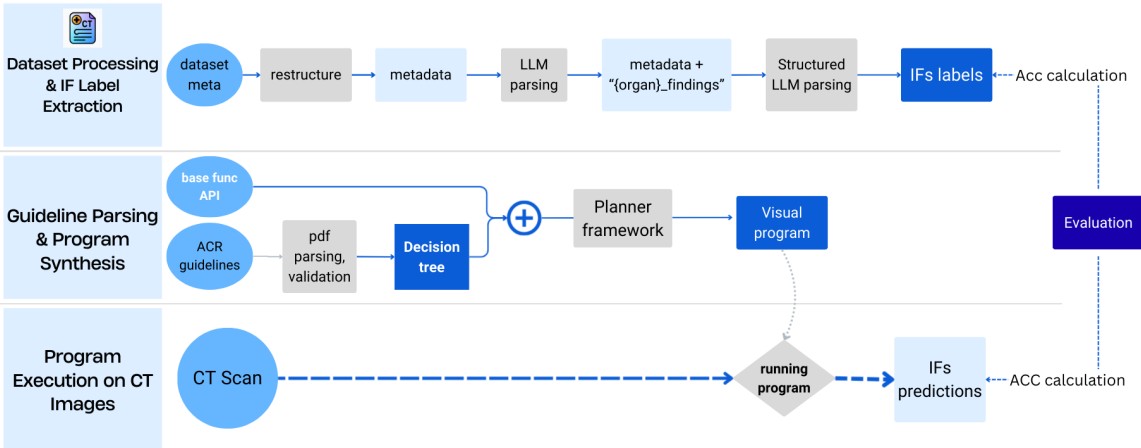

Figure 1: Overview of INFORM-CT pipeline. The framework consists of three components: (i) *Dataset Processing & IF Label Extraction* (see Section 3.1.1): structured metadata is derived from radiology reports through LLM-based parsing, enabling the extraction of organ-specific incidental findings (IFs) labels; (ii) *Guideline Parsing & Program Synthesis* (see Section 2) medical guideline are parsed into decision-tree structures, which are combined with a set of predefined base functions to generate an executable inspection program via a planner–executor architecture; (iii) *Program Execution*: The program operates on CT scans, invoking base functions to produce IF predictions. These predictions are evaluated against IFs labels to compute accuracy measures.

in this inspection. For instance, a single program might include the detection of a tumor mask, the calculation of its diameter (in mm), the measurement of its border thickness, tumor gray-level evaluation (in Hounsfield units), and the presence of higher-level attributes assessed by a CLIP classifier — all in addition to the logical options inherent in the Python script itself. These complex requirements demand extensions of existing plan-and-execute methods into more sophisticated programs with an expanded set of base functions.

A representative example of a synthesized program derived from the ACR liver guidelines is shown in Algorithm 1 (Appendix A), illustrating the type of clinical logic produced by our planner–executor framework.

### 2.2.1. BASE FUNCTIONS.

The base functions are built on existing methods, models, and detectors for segmentation and detection of CT organs, such as abdomen CT segmentation models, abdomen CLIP models, and image processing procedures. These functions include:

- *Organ Segmentation*: Segmenting organs in the CT scan. Based on TotalSegmentor (Wasserthal et al., 2023), and nnUNet (Isensee et al., 2021) frameworks. These

Figure 2: Guideline Parsing Process Demonstration. The left panel presents a section of the original guideline document in PDF format, while the right panel displays the corresponding parsed JSON structure, which encapsulates key nodes, verifications, and risk assessments, facilitating integration into the planner-executor framework.

include multiple different segmentation models that cover a wide range of tasks, including organ and tumor segmentation in the abdomen.

- *Mass and Tumor Segmentation*: Detecting and segmenting masses and tumors. Based on (Isensee et al., 2021; Wasserthal et al., 2023) as well.

- *Measuring tumor diameter*: An image processing procedure to measure the diameter (in cm or mm) of a tumor based on a mask of pixels and metadata from the CT resolution. Includes a few estimation methods.

- *Measuring gray-level intensity*: An image processing procedure to measure the gray-level intensity (HU) of a tumor based on a mask of pixels and metadata from the CT scan file.

- *Measuring border thickness*: Measuring the thickness of organ or lesion borders using the Hausdorff distance.

- *Labeler*: A labeler module is integrated to automate the classification of higher-level fine-grained attributes using a vision-language model. For example, the labeler can tag a lesion as "benign", "suspicious", or "flash-filling" according to a list of sub-features. Our labeler is implemented using the MERLIN model (Blankemeier et al., 2024), which is currently the state-of-the-art 3D model for abdominal CT. It was trained on paired 3D CT volumes and corresponding text reports, enabling it to generate accurate labels for segmented regions on these scans.

2.2.2. INCIDENTAL FINDINGS CODE GENERATION.

Code generation models such as (Gupta and Kembhavi, 2023; Surís et al., 2023) have demonstrated the successful use of creating programs as a description of complex decision-

making and analysis processes. However, clinical detection of incidental findings in abdominal CT scans involves a more challenging task. This process must account for multiple critical factors that are not typically used for normal images, including computation of size and grey-level intensity in specified regions, considering scan details such as contrast phase, and often incorporating patient medical history. These complexities necessitate a robust and adaptable code generation approach to ensure accurate and efficient analysis.

To generate a code representation of each incidental findings management procedure, we provided a detailed description of the API available for each base function, along with simple examples that demonstrate their proper usage. In addition, we included a comprehensive overview of the problem, the clinical pipeline, the parsed tree from Sec. 2.1 as well as other relevant details to effectively instruct the code generation process.

The generation process works in an interactive manner, involving a multi-turn conversation with the LLM. The steps are as follows:

1. *Initial Draft Generation:* The agent generates a preliminary draft of the program out of the decision tree.

2. *Execution and Feedback:* After executing the draft, the system generates feedback and evaluates a STOP criterion to decide whether to continue. This criterion assesses both syntactic correctness and semantic validity of the generated code.

3. *Iterative Refinement:* If the STOP criterion is not met, another call to the LLM is made to regenerate the code based on the feedback. The method then returns to step 2, iterating through these steps until the final program is produced.

## 3. Experiments

### 3.1. Multi-Organ Benchmark from Clinical Abdominal CT

To simulate the real-world management of incidental findings, where multiple organs need to be checked according to various guidelines (in PDFs) and adhering to hospital protocols, we conducted a benchmark using American College of Radiology (ACR) guidelines for three different organs: Liver, Pancreas, and Kidney. The experiments are based on our internal dataset, which consists of a large set of abdominal CT scans paired with radiology reports. This dataset was used to develop a procedure for collecting test data for our method. The data includes thousands of abdominal CT scans from 6,366 unique patients. The scans were made in various phases, including venous and arterial phases.

For liver scans, we restricted our investigation to venous phase CT scans, allowing existing segmentation models to effectively detect lesions. To ensure a balanced set of recommendations, we included scans both with and without liver lesion detections. This approach allowed us to cover the different paths in the decision tree comprehensively. Consequently, we included scans that were conducted for liver inspection, where liver lesions are more likely to be found. Ultimately, we gathered 168 scans, providing a good balance of the possible decision tree paths.

We applied the same type of filtering for the pancreas and kidney. For the pancreas, we selected venous phase CT scans, while for the kidney, we chose arterial phase CT scans.

This approach ensured that lesions in these organs were detectable by existing segmentation models, allowing us to create a balanced and comprehensive dataset for each organ. Specifically, we gathered 168 scans for the liver, 188 scans for the pancreas, and 98 scans for the kidney.

To ensure our method adhered to established clinical guidelines, we selected guidelines from the ACR website and collected PDFs for each organ. We then used the parsed tree procedure described in Section 2.1 to parse these guidelines. This process involved converting the PDF text and figures into JSON formats, which included structured information such as checks, detections, measurements, and recommendations.

### 3.1.1. EXTRACTING "CORRECT" RECOMMENDATIONS FROM REPORTS.

For each scan, our method aims to generate recommendations based on guidelines for the management of incidental findings in a specific organ. To evaluate the predicted recommendations, we built a procedure to also obtain "correct" recommendations extracted from reports. First, we note that the radiologist's report for each scan includes detailed observations and patient background information, which can be used to infer the recommendation and its explanation as reflected by a trajectory in the parsed tree. Next, we generated a list of all possible paths in the decision tree by traversing it, with each path representing a sequence of checks and decisions leading to a specific recommendation. We then used an LLM (GPT-4o, (OpenAI, 2023)) to review the radiology report and select the best tree path that matches the report. This selected path is considered the "correct" recommendation for the scan (the leaf includes the recommendation, while the rest of the path can be considered its "explanation"), which we use to test both baseline and our model.

### 3.2. Evaluation of INFORM-CT

#### 3.2.1. IMPLEMENTATION DETAILS.

We utilized the CLAUDE 3.5 LLM (Anthropic, 2024) to intelligently translate the logic described in the guidelines into code. For example, the guidelines might specify that a mass is "Homogeneous (thin or imperceptible wall, no mural nodule, septa, or calcification)." This description needs to be converted into code that operates the base functions. The logic of "or" and "not", as well as the computation of attributes such as "thin", "thick", and "calcification" are all handled by the base functions and orchestrated by the program. We implemented most of the image processing functions using standard Python libraries. For running MERLIN as a labeler to obtain higher-level attributes, we computed the cosine similarity of the entire 3D scan matched to a set of labels representing all potential attribute values. We were limited in this implementation by the variety of available strong segmentation models for abdominal organ lesions. All segmentation models were implemented using nnUNET and taken from its GitHub repository (Isensee et al., 2021; Wasserthal et al., 2023).

All programs were automatically generated from a PDF containing guidelines. Our experiments included guidelines from the ACR, but any adjustment of these guidelines according to specific hospital protocols, as well as guidelines from different radiology organizations (e.g., the ESR - European Society of Radiology), can be accommodated.

| Abdominal Organ | Random prediction | Pure MERLIN | INFORM-CT |
|:---:|:---:|:---:|:---:|
| Liver | 10.0 / – | 12.5 / 14.33 | 63.09 / 61.38 |
| Renal | 16.67 / – | 48.8 / 14.0 | 60.0 / 61.0 |
| Pancreas | 7.14 / – | 11.17/ 12.96 | 41.48 / 46.32 |

Table 1: Accuracy (left, in percentage) and weighted F1 scores (right) for predicting the correct recommendation in managing incidental findings across selected organs using the INFORM-CT and MERLIN models.

### 3.2.2. Baseline Evaluation Using MERLIN Model.

Our model was evaluated against the MERLIN baseline (Blankemeier et al., 2024), a Vision Language Model similar to CLIP. To evaluate MERLIN, we listed all the possible paths in the decision tree, concatenating the text in the nodes. We also included the patient background information (such as age and risk factor), as it was provided to INFORM-CT, to ensure a fair comparison. For each decision path combined with the background information, we computed the cosine similarity and selected the path with the highest score as the prediction of the MERLIN model.

### 3.2.3. Analysis.

The results of matching the recommendation predictions of our models, as well as the MERLIN model (as mentioned above), are shown in Table 1. Figure 3 illustrates the process, showing link to the guideline PDF, pieces from the generated code, the execution of the code via base functions, and the execution output for liver and kidney (renal) incidental findings management over two sample scans. The results indicate that our method can effectively handle the automatic management of incidental findings for different abdominal organs. The accuracy of the final recommendation predictions is relatively high, typically much higher than applying a pure VLM approach ("Pure Merlin") on a real-world clinical benchmark. We also evaluated the correctness of decisions made along the way, namely the path in the decision tree that yielded the recommendation. This is the explainable part of our model, and this evaluation sheds light on how explainable the model is and how well it matches the correct explanation computed from the report (as explained in Section 3.1). The results, as shown in Table 2, indicate that the model explanation matches those provided in the report for the majority of cases, while MERLIN provides limited explanatory capability.

Finally, we turn to assess the contribution of the internal components of the model on performance. Specifically, we evaluate the contribution of the segmentation base functions through an ablation study. In Table 2, we present the recommendation accuracy (in percentage) of an ablated INFORM-CT model in which segmentation tasks are converted to text and are also performed by MERLIN. Comparing this ablated model to the full INFORM-CT reveals that the segmentation component is critical to the success of the model and cannot be replaced by the VLM. However, the VLM and image processing routines are also crucial components of INFORM-CT, leading to the conclusion that the whole is greater than the sum of its parts.

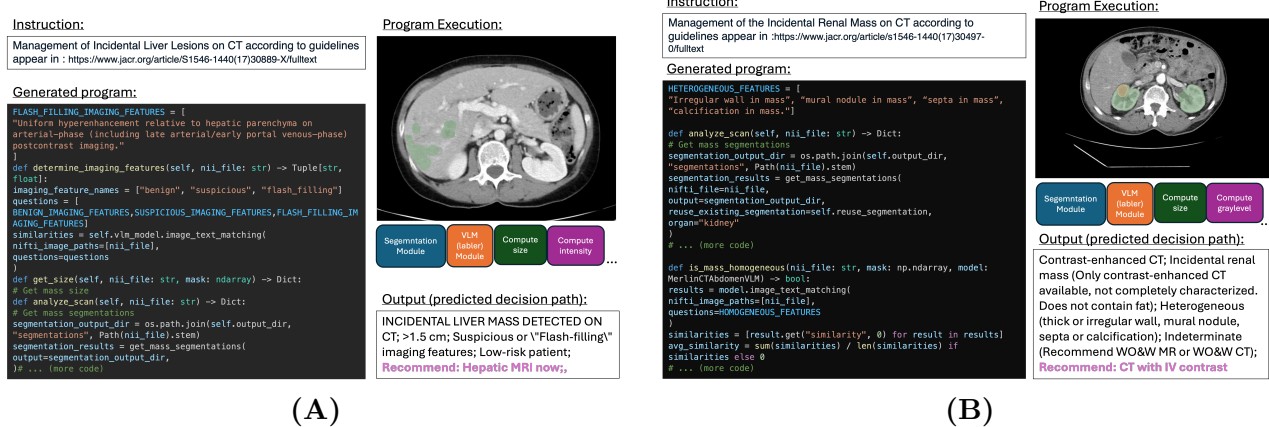

Figure 3: Predictions of the INFORM-CT model for scans, adhering to ACR guidelines for the management of incidental findings. Process and results are shown for the liver (A), renal (B). Selected tree trajectory is shown as output, and final recommendation is marked magenta color.

Table 2: Additional evaluation for the incidental finding management of the liver. The explanatory part of the model is shown on the left, displaying the accuracy of the decision trajectory for obtaining the final recommendation matched to the reasons provided in the report. On the right is a comparison of the full and ablated model, where the segmentation base routine is removed and replaced by the VLM.

| Explanation Evaluation | | |
|---|---|---|
| | INFORM-CT | Pure MERLIN |
| Acc | 54.76 | 4.76 |

| Recommendation Evaluation | | |
|---|---|---|
| | Full | Ablated |
| Acc | 63.09 | 20.45 |

## 4. Discussion

In this paper, we have demonstrated the effectiveness of the INFORM-CT agentic framework in managing incidental findings on abdominal CT scans by leveraging advanced LLM, VLM, and segmentation models. While INFORM-CT can effectively handle the logic and image processing operations needed for incidental findings, it is still limited by the capabilities of the underlying segmentation and VLM models used as base functions. Specifically, we expect that a VLM capable of better labeling fine details against local scan regions will further improve recommendation performance.

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

## Appendix A. Example Generated Programs

Algorithm 1 provides an illustrative example of the type of clinical logic executed by INFORM-CT. Starting from the segmented liver masses, the algorithm computes lesion-level attributes—including diameter, radiological features, and patient-specific risk factors—and applies the decision rules derived from the ACR guidelines to produce per-lesion recommendations. These are then aggregated into a patient-level follow-up recommendation.

---

**Algorithm 1:** Assessment of Liver Lesions

---

*This illustrative pseudocode shows the structure of a synthesized program generated by the planner–executor framework for the liver ACR guidelines.*

**Input:** $x$: abdominal CT scan
**Input:** $b$: patient background / clinical data
**Output:** $rec$: patient-level follow-up recommendation
$\mathcal{M} \leftarrow$ `mass_segmentator`($x$, *organ="liver"*)
$R \leftarrow [\,]$       `// list of mass-level recommendations`
**foreach** $m \in \mathcal{M}$ **do**
  $d \leftarrow$ `calc_mass_diameter_cm`($m$, $x$)     `// mass diameter in cm`
  $r \leftarrow$ `assess_patient_liver_risk`($b$)    `// patient-level risk (Low / High)`
  **if** $d \leq 1.0$ **then**
    **if** $r = Low$ **then**
      $r_m \leftarrow$ "Benign; no further follow-up."
    **else**
      $r_m \leftarrow$ "Liver MRI in 3–6 months."
  **else if** $1.0 < d \leq 1.5$ **then**
    $\phi_m \leftarrow$ `calc_mass_imaging_features`($m$, $x$)  `// lesion's imaging features`
    **if** $\phi_m = suspicious$ **then**
      *Further logic...*
    **else**
      *Further other logic...*
  **else**
    "Further logic for large lesions..."       `// larger than 1.5 cm`
  append $r_m$ to $R$
$rec \leftarrow$ `agg_recommendations`($R$)     `// aggregate into patient-level`
`recommendation`
**return** $rec$

---

While simplified for clarity, this example captures the core reasoning steps synthesized automatically by the planner–executor framework, which generates similar structured programs for all organs and guideline pathways.

**Pipeline failures.** We observe failure cases where segmentation is correct, but the downstream interpretation of imaging features is incorrect. For example, in one case the system correctly segmented three hepatic masses with sizes of approximately 2.0 cm, 0.7 cm, and 0.4 cm. According to the ground truth, the imaging appearance was benign

and did not warrant additional follow-up. However, the model classified the findings as having "suspicious features," which triggered a more conservative guideline pathway and an incorrect recommendation for further imaging. These failure could happen even though the question was decomposed into a small building blocks

**Planning strategies.** Here are a few examples demonstrating the planner solution to complex decisions required by the guidelines. In Algorithm 2, we illustrate the planner's decomposition mechanism, where a top high-level concept ("high-risk features") is expanded into a set of down atomic, interpretable VLM queries (e.g., mural nodules, solid components, ductal dilation). This produces per-factor confidence scores that can be explicitly aggregated according to clinical rules, yielding an interpretable risk assessment.

---

**Algorithm 2:** Planner Decomposition of "High-Risk Features" into Atomic VLM Queries

---

**Input:** CT scan $\mathcal{I}$, VLM $\mathcal{V}$, threshold $\tau$
**Output:** Per-factor evidence $E$, aggregated flag $HR$
Function(VLMYesNo($\mathcal{I}, \mathcal{V}, q$)) $p \leftarrow \mathcal{V}$.IMAGETEXTMATCHING($\mathcal{I}, [q]$)
**return** $p$

$\mathcal{F} \leftarrow \{q_1$: "mural nodule present",
$q_2$: "solid component present",
$q_3$: "main pancreatic duct dilated ($\geq$ 10mm)",
$q_4$: "abrupt duct caliber change with distal atrophy"$\}$
**foreach** $(k, q_k) \in \mathcal{F}$ **do**
    $p_k \leftarrow$ VLMYesNo($\mathcal{I}, \mathcal{V}, q_k$)
    $E[k] \leftarrow (p_k,\ p_k > \tau)$                 // (confidence, boolean)
$HR \leftarrow$ AggregateRisk($E$)     // e.g., OR over booleans, or weighted rule
**return** $(E, HR)$

---

Complementarily, The pipeline sometimes has to fuse global and local information, not only classification of mass-level properties. Algorithm 3 demonstrates the use of the VLM to answer a contextual anatomical question—main pancreatic duct (MPD) communication—that is not directly tied to the segmented lesion.

---

**Algorithm 3:** VLM Query for MPD Communication

---

**Input:** CT scan $\mathcal{I}$, VLM $\mathcal{V}$, threshold $\tau$
**Output:** MPD communication flag $C$, confidence score $s$
Function(ScorePrompt($\mathcal{I}, \mathcal{V}, q$)) **return** $\mathcal{V}$.IMAGETEXTMATCHING($\mathcal{I}, [q]$)

$q^+ \leftarrow$ "There is evidence the cyst communicates with the main pancreatic duct (MPD)."
$q^- \leftarrow$ "There is no evidence of MPD communication with the cyst."
$p^+ \leftarrow$ ScorePrompt($\mathcal{I}, \mathcal{V}, q^+$)
$p^- \leftarrow$ ScorePrompt($\mathcal{I}, \mathcal{V}, q^-$)
$s \leftarrow p^+ - p^-$                           // consistency margin
$C \leftarrow (s > \tau)$
**return** $(C, s)$

---

