# OpenReview forum: "INFORM-CT: INtegrating LLMs and VLMs FOR Incidental Findings Management in Abdominal CT"
_MIDL.io/2026/Conference — MIDL 2026 Poster_

### Official Review · Reviewer_S54p · 2025-12-29

**Confidence:** 4
**Preliminary Rating:** 4
**Final Rating:** 4

**Summary:**

This paper introduces INFORM-CT, an agentic framework that uses LM, VLM, computer vision routines to automate the detection of incidental findings in abdominal CT. The authors uses LM to use clinical guidelines as context and translate that to python programs and then execute these programs that call vision/vlm subroutines to gather evidence and provide the final guidelines. The authors evaluated their approach on multi-organ abdominal CT benchmark and show substantial gains over pure VLM detector.

**Strengths:**

- This paper shows promise in using agent framework to automate clinical workflow end to end. It is quite flexible in that the detection relies on clinical guidelines that can be updated/altered and can be adapted to varying guidelines using LMs. It shows a lot of promise in automating real-world radiology workflows.
- The proposed method provide very substantial improvement in incidental finding detection than baseline. This is very promising. Some discussion on what contributed to this performance gain would be helpful.
- The proposed method is also quite interpretable, since it could pinpoint the exact criteria for the tumor that supports/refutes its classification. This would be helpful for building trust with end-users.

**Weaknesses:**

- The paper is evaluated from an internal dataset that is relatively small, limiting its conclusions about generalizability/robustness.
- The paper lacks comparison to baseline that fixes to a reasonable pipeline, e.g., segmentation then detect incidental findings. This could clarify how much benefit comes from having LM synthesizing programs. This is a pretty important ablation.
- Some qualitative analysis of failure modes (e.g., segmentation error, program synthesis failures, etc.) would be helpful to understand the state of the entire agentic system.
- It would be interesting if the authors would provide example workflow for a few cases for the same and/or different guidelines. It's probably interesting to know the solution space generated by the LM planner. Does it always follow segmentation -> measuring statistics of the tumor -> detection? Does some tools more useful for certain types of organ/findings than others? They are super interesting questions to think about.
- Related to previous, it would also be interesting if the authors could provide more comprehensive ablations that let the reader understand a bit more on relative contribution of the different tools. Currently we know that segmentation is a very important.

**Detailed Comments:**

n/a

**Justification Of Final Rating:**

The rebuttal addressed my concerns. I also find the findings that LM generated diverse solution spaces that the author's supplied in the revised paper. I'll keep the rating of weak accept. Thank you!!

**Justification Of The Preliminary Rating:**

The paper addresses an important and practical radiology problem and proposes a flexible, guideline-driven agentic framework that meaningfully advances beyond pure VLM-based approaches. The integration of LMs for program synthesis with vision and image-processing tools is well motivated, interpretable, and shows substantial empirical gains on a realistic multi-organ benchmark. While the evaluation is limited to a small internal dataset and lacks some key ablations and failure analyses, the paper does provide solid contribution. I recommend weak accept pending stronger validation and deeper analysis.

**Questions To Address In The Rebuttal:**

see weaknesses

---

> ### Author Response · Authors · 2026-01-25
> **Response to reviewer S54p**
>
> We thank the reviewer for the thoughtful evaluation of our work and for highlighting the interpretability of the proposed framework. We are encouraged by the reviewers' consistent recognition of INFORM-CT's novelty and clinical motivation. Across the feedback, questions regarding generalization and failure mode analysis emerged as recurring themes. In the responses below and in the revised manuscript, we address these points through an error breakdown and a discussion of the framework’s modular extensibility.
>
>
> >The paper is evaluated from an internal dataset that is relatively small, limiting its conclusions about generalizability/robustness.
>
> We acknowledge that the current evaluation on our internal dataset limits conclusions regarding population-level generalization and robustness. However, the LM-based planner in INFORM-CT composes tool calls based solely on medical guidelines—without learning from CT images or reports—making its generalization primarily dependent on the underlying perception components (segmentation, VLMs) rather than dataset size. The evaluation thus focuses on characterizing realistic system behavior and failure modes under clinical constraints, though we note that these failure modes are demonstrated only on this dataset and may differ at population scale.
>
> > The paper lacks a comparison to a baseline that fixes to a reasonable pipeline, e.g., segmentation, then detects incidental findings. This could clarify how much benefit comes from having LM synthesizing programs. This is a pretty important ablation.
>
>
> To examine the role of LM-based program synthesis, we include a fixed two-stage baseline consisting of organ and lesion segmentation followed by direct classification of incidental findings from the segmented regions, without adaptive program generation or conditional logic. While this pipeline represents a reasonable static baseline, many guideline-driven assessments in abdominal CT require conditional reasoning over multiple factors (e.g., lesion properties, patient risk factors) that cannot be expressed by a single fixed decision path. Consistent with this observation, our ablation results show a performance degradation when adaptive program synthesis is removed, indicating that the observed gains arise from guideline-conditioned decision logic rather than from segmentation or classification alone.
>
> > Some qualitative analysis of failure modes (e.g., segmentation error, program synthesis failures, etc.) would be helpful to understand the state of the entire agentic system.
>
> Typical failure cases in the INFORM-CT pipeline occur when the labeler attempts to estimate the mass-level imaging features. These features describe fine-grained visual properties of an organ region, which current SOTA multimodal models still find challenging, even with careful visual prompting. In such cases, the labeler may misclassify the presence or absence of an imaging feature (e.g., “thick septa”), leading to the assignment of incorrect high-level descriptors (e.g., benign vs. suspicious). Consequently, the final follow-up recommendation generated by the planner may be incorrect, despite the planner correctly decomposing the imaging features into specific, clinically interpretable labeling queries.
>
> Segmentation errors represent the second failure mode—less frequent than labeler issues, particularly when operating within appropriate scan phases. While reliable relative to the labeler component, accurate segmentation remains important, as competing multimodal models such as Merlin cannot perform it end-to-end.
>
>
> > It would be interesting if the authors would provide example workflow for a few cases for the same and/or different guidelines. It's probably interesting to know the solution space generated by the LM planner. Does it always follow segmentation -> measuring statistics of the tumor -> detection? Does some tools more useful for certain types of organ/findings than others? They are super interesting questions to think about. Related to previous, it would also be interesting if the authors could provide more comprehensive ablations that let the reader understand a bit more on relative contribution of the different tools. Currently we know that segmentation is a very important.
>
>
> We evaluated the solution space induced by the LM planner across clinical guidelines and observed diverse, guideline-specific workflows rather than a fixed procedural pattern. In several cases, now included in the revised manuscript, decision logic is driven by patient risk factors and clinical history, without requiring explicit segmentation or quantitative tumor measurements. These findings indicate that the planner faithfully instantiates guideline-dependent logic and that the resulting workflows cannot be reduced to a simple segmentation–detection pipeline.

---

### Official Review · Reviewer_SiVd · 2025-12-31

**Confidence:** 4
**Preliminary Rating:** 4
**Final Rating:** 4

**Summary:**

This paper addresses the challenge of managing incidental findings on abdominal CT scans. Incidental findings are extremely common and must be reported in accordance with established guidelines. The authors propose INFORM-CT, a novel framework that integrates Large Language Models (LLMs) with Vision–Language Models (VLMs) in a plan-and-execute agentic architecture. The goal is to automatically detect, characterize, and recommend management for incidental findings in abdominal CT scans, adhering strictly to clinical guidelines without requiring direct radiologist input.

**Strengths:**

1. This paper tackles a new research direction by combining an LLM (for reasoning and code-generation) with medical imaging models in a unified agent. This is the first application of a plan-and-execute LLM agent for CT diagnosis tasks.
2. A major strength is that the system’s decisions are grounded in established medical guidelines (e.g. ACR recommendations). The approach produces not only a recommendation but an explanation path following a guideline flowchart. This yields an interpretable decision trail that clinicians can review, unlike a black-box neural network.
3. This paper provides a clear breakdown of the planner–executor architecture and a thorough description of each base function.

**Weaknesses:**

1. While this paper shows promising results, it is restricted to three organs (liver, kidney, pancreas) and a relatively small test set (on the order of ~100–200 cases per organ).
2. The INFORM-CT pipeline is quite complex, relying on multiple AI components: an LLM for parsing guidelines, another LLM for code-generation, several segmentation networks, and a VLM for attribute labeling. The overall success depends on all of these working well.
3. While INFORM-CT outperforms the baseline, the absolute accuracy is moderate.

**Detailed Comments:**

1. While this paper shows promising results, it is restricted to three organs (liver, kidney, pancreas) and a relatively small test set (on the order of ~100–200 cases per organ). This raises concerns about the framework’s generality. The method is advertised as a pipeline for the “entire abdominal region”, but important organs like the adrenal or spleen are not included, likely due to a lack of available segmentation models or guidelines used.
2. The INFORM-CT pipeline is quite complex, relying on multiple AI components: an LLM for parsing guidelines, another LLM for code-generation, several segmentation networks, and a VLM for attribute labeling. The overall success depends on all of these working well. If any component underperforms, the final recommendation may be wrong. This complexity may also make it challenging to deploy and maintain in practice compared to a single, end-to-end model.
3. While INFORM-CT outperforms the baseline, the absolute accuracy is moderate. It’s not fully discussed what kinds of mistakes are being made or how critical they are.

**Justification Of Final Rating:**

While the current performance is not yet at a level for clinical deployment (as the authors admit, ~80% is needed), the work opens a valuable new research direction. I will maintain my score of 4. Thanks to all authors' hard work. Good luck!

**Justification Of The Preliminary Rating:**

This work presents a creative and original solution to a significant problem in medical imaging. The integration of LLM planning with vision models is cutting-edge and, to my knowledge, novel in the radiology domain.

**Questions To Address In The Rebuttal:**

1. How easily can INFORM-CT be extended to other organs or updated guidelines? If only three organs are handled currently, what are the barriers to adding more?
2. What are the common failure modes for cases where INFORM-CT’s recommendation was wrong?
3. Given the current accuracy (~60%), how do you envision this tool being used by radiologists? Would it be a second reader who suggests recommendations for review? What level of accuracy would be needed for it to be trusted in practice, and how might future improvements reach that?

---

> ### Author Response · Authors · 2026-01-25
> **Response to reviewer SiVd**
>
> We thank the reviewer for the detailed and constructive review and insightful comments. We appreciate your positive evaluation, consistent with that of the other reviewers. Below we address the points raised, with clarifications regarding evaluation design and clinical relevance.
>
> > How easily can INFORM-CT be extended to other organs or updated guidelines? If only three organs are handled currently, what are the barriers to adding more?
>
> Our framework is organ and guideline-agnostic. The LLM can interpret clinical guidelines and generate clinical diagnosis programs using the appropriate tools. However, extending the framework to additional organs beyond the three examined requires the following:
> Data which includes findings in the specific organ, along with study reports (that would be translated into mass level annotations in our annotation pipeline). In our internal dataset, other organs were mostly benign, and lesions and masses on these organs were infrequent.
> Available Open-access models for detection and segmentation, and classification of masses in the additional organ.
> We hope for further developments in mass level annotation datasets [such as, AbdomenAtlas 3.0] that would support more accurate mass level VQA, classification, and segmentation models.
>
>
> > What are the common failure modes for cases where INFORM-CT’s recommendation was wrong?
>
> Typical failure cases in the INFORM-CT pipeline occur when the labeler tries to estimate the mass level imaging features. These features describe fine-grained visual properties of an organ region (e.g., septations), which current state-of-the-art multimodal models such as RadGPT and f-VLM still find challenging, even with careful visual prompting. In such cases, the labeler may misclassify the presence or absence imaging feature (e.g., “thick septa”), leading to the assignment of incorrect high-level descriptors (e.g., benign vs. suspicious). Consequently, the final follow-up recommendation generated by the planner may be incorrect, despite the planner correctly decomposing the imaging features into specific, clinically interpretable labeling queries.
>
> > Given the current accuracy (~60%), how do you envision this tool being used by radiologists? Would it be a second reader who suggests recommendations for review? What level of accuracy would be needed for it to be trusted in practice, and how might future improvements reach that?
>
> We propose INFORM-CT as an assistive second-reader system that proposes guideline-based recommendations for radiologists to review during scan analysis (could be processed offline at a compute center before inspection). Its role would be to surface candidate follow-up recommendations and highlight actionable incidental findings with explicit interpretable reasoning, enabling radiologists to rapidly verify or modify suggestions.​
> The path to clinical deployment requires improved performance approaching levels comparable to clinician agreement (with clinicians indicating ~80% success rate would be suitable), plus stratified evaluation by organ and finding severity. Future improvements will leverage stronger abdominal CT perception models (segmentation, VLMs), leading to straightforward integration of new tools into the modular INFORM-CT framework, as well as from lightweight, targeted training or calibration (e.g., prompt policies or tool-selection thresholds) that does not rely on heavy end-to-end fine-tuning. In addition, the agentic formulation enables human feedback at intermediate nodes of the decision process (e.g., synthesized programs, explicit queries, and clinical measurements), which can be used both to intervene at specific reasoning steps and to iteratively refine the system over time, rather than only at the final output.

---

### Official Review · Reviewer_dypR · 2026-01-10

**Confidence:** 4
**Preliminary Rating:** 4
**Final Rating:** 4

**Summary:**

This paper introduces INFORM-CT, an automated framework for handling incidental findings in abdominal CT by combining large language models, vision-language models, and classical image processing in a planner–executor setup. Clinical guidelines, provided as PDFs, are first converted into structured decision trees, which an LLM then translates into executable Python code that calls segmentation models, measurement routines, and VLM-based labeling modules to produce guideline-consistent recommendations. Experiments on internal abdominal CT datasets covering the liver, pancreas, and kidney show that the proposed system clearly outperforms a baseline relying solely on VLMs for predicting appropriate clinical recommendations. In addition, the system outputs an explicit record of intermediate decisions aligned with the underlying guidelines, which could be valuable in clinical practice. Overall, the results suggest that protocol-driven approaches to automated image analysis have meaningful potential for radiology applications.

**Strengths:**

The main strength of this paper is its systems-level integration of LLMs, VLMs, and established medical imaging components into a coherent, clinically motivated pipeline. Rather than proposing yet another end-to-end black-box model, the authors emphasize interpretability, which are critical for clinical use. The planner-executor architecture is in line with the structured logic of radiology guidelines and provides a flexible mechanism for incorporating new protocols. The experimental results show a potential improvement over a strong VLM baseline. Overall, the paper is clearly written and well structured, and it has meaningful potential impact for the medical imaging filed.

**Weaknesses:**

A notable limitation of this work is the reliance on an internal dataset and LLM-generated labels for evaluation, which raises concerns about potential bias and also limits external validation. The experimental evaluation is also fairly limited, both in the number of organs studied and in the range of clinical scenarios considered, making it difficult to judge how robust the proposed framework would be in broader clinical settings. Although the authors position the approach as general-purpose, its performance appears to depend strongly on the availability and quality of pretrained segmentation models, which may not transfer well across institutions or imaging protocols. In addition, the evaluation primarily reports aggregate accuracy metrics, with little analysis of clinically meaningful failure modes, such as incorrect follow-up recommendations that could have real consequences for patient care. Finally, while the planner-executor design is interesting, its incremental novelty relative to existing modular or rule-based clinical decision systems is not entirely clear and would benefit from more explicit discussion and comparative experiments.

**Detailed Comments:**

1.It would be useful if the authors could clarify how ambiguities or conflicting statements in the guideline PDFs are handled during the parsing process.
2.Reporting basic runtime statistics for guideline parsing, code generation, and execution would help readers better assess the practicality of the proposed system.
3.Including a small number of qualitative examples of failure cases or incorrect recommendations could further strengthen the experimental evaluation.
4.The discussion could be expanded to more clearly explain how guideline updates would be incorporated in practice after deployment.
5.Some additional clarification on how patient background information is encoded and utilized by the system would improve the overall clarity of the paper.

**Justification Of Final Rating:**

Thanks to the authors' efforts. The rebuttal addressed my concerns, and the revised paper is clearer overall. While the approach is not yet ready in practice, the ideas are promising for future work. I will keep my rating as weak accept.

**Justification Of The Preliminary Rating:**

I lean toward a weak accept. The paper proposes a thoughtful and clinically motivated framework that moves beyond incremental model tweaks and instead tackles workflow-level issues in radiology. The combination of guideline parsing, program synthesis, and explainable execution is novel in the context of abdominal CT and incidental findings management. There are valid concerns around the evaluation setup, dataset accessibility, and the overall scope of the experiments, but these issues do not fundamentally detract from the core technical contribution. With clearer validation and a more candid discussion of limitations, the work could help shape future research on agent-based systems for medical imaging, which in my view supports acceptance despite the noted weaknesses.

**Questions To Address In The Rebuttal:**

How sensitive are the results to errors in the LLM-based extraction of the "correct" recommendations from the reports?

Is there any evidence that the planner-executor framework would generalize beyond the three organs studied here?

How does the system behave when the segmentation quality is poor or when lesions are missed altogether?

How do the computational costs compare with a standard, end-to-end VLM baseline?

In a real clinical setting, how would this system be validated and monitored over time?

---

> ### Author Response · Authors · 2026-01-25
> **Response to reviewer dypR**
>
> We thank the reviewer for their constructive and valuable feedback. We are glad for the positive evaluation, which highlights similar aspects related to generalization to additional organs and failure modes. We have carefully analyzed these points and provide detailed clarifications (along with updated paper revision) in the responses below.
>
> > How sensitive are the results to errors in the LLM-based extraction of the "correct" recommendations from the reports?
>
> To reduce labeling errors, we designed a careful annotation pipeline with a programmatically validated JSON schema encoding detailed mass attributes (e.g., location, size, imaging features, key report phrases). The pipeline operated agentically, iteratively validating JSON outputs against the raw text report until semantic correctness. Complex scenarios like multiple lesions in one organ or size changes across follow-up studies were explicitly tested and supported.
>
> > Is there any evidence that the planner-executor framework would generalize beyond the three organs studied here?
>
> For other organs beyond the three studied in this work, the parsing and the planning components of the INFORM-CT pipeline remain applicable, as they primarily rely on document analysis capabilities. However, in our internal dataset, other organs were mostly benign, and lesions and masses on these organs were relatively rare. To demonstrate generalization capabilities to additional organs, the system would require larger open-access CT datasets with sufficient number of masses, paired with radiology reports, and detailed mass-level annotations (e.g., attributes, segmentation masks) for these organs.
>
> > In a real clinical setting, how would this system be validated and monitored over time?
> > The discussion could be expanded to more clearly explain how guideline updates would be incorporated in practice after deployment.
>
> In clinical practice, INFORM-CT would be validated prospectively against radiologist consensus on guideline adherence, monitored via audit logs of synthesized programs, intermediate tool calls, and final recommendations. Radiologists can inspect the explicit decision tree at each step, providing feedback that triggers case review or retraining.
> Deployment runs between CT acquisition and radiologist interpretation on central compute infrastructure. Guideline updates are automatic: the planner re-parses revised textual protocols to synthesize new programs without redesign. With high annual abdominal CT volume (~1M exams in the U.S.), ample data exists for continuous validation and adaptation across hospitals.
> > Some additional clarification on how patient background information is encoded and utilized by the system would improve the overall clarity of the paper.
>
> Patient factors are incorporated through structured extraction of key-value fields (e.g., age, comorbidities) and free-text identification of guideline-defined risks such as smoking. These inputs guide the planner to ensure guideline-compliant risk stratification and follow-up, for example prioritizing young patients under 35 with suspicious masses.
>
> > Reporting basic runtime statistics for guideline parsing, code generation, and execution would help readers better assess the practicality of the proposed system
>
> > How do the computational costs compare with a standard, end-to-end VLM baseline?
>
> In our current implementation, end-to-end inference pipeline execution typically completes in a few minutes per case, where the main computational cost comes from organ and lesion segmentation. The VLM labeler step of the execution is lighter, taking ~20-30 s per scan (The heavy part is 3D CT volume encoding). Guideline parsing and code generation are comparatively lightweight, each requiring only a small number of agent iterations. The planning stage involves a limited number of agent iterations over the decision tree and base-function API, resulting in on the order of a few minutes per case.
>
> > How does the system behave when the segmentation quality is poor or when lesions are missed altogether?
> > Including a small number of qualitative examples of failure cases or incorrect recommendations could further strengthen the experimental evaluation.
>
> We observe failure cases where segmentation is correct, but the downstream interpretation of imaging features is incorrect. For example, in one case the system correctly segmented three hepatic masses with sizes of approximately 2.0 cm, 0.7 cm, and 0.4 cm. According to the ground truth, the imaging appearance was benign and did not warrant additional follow-up. However, the model classified the findings as having “suspicious features,” which triggered a more conservative guideline pathway and an incorrect recommendation for further imaging. This example illustrates a failure mode in which feature interpretation and risk assessment are miscalibrated, rather than errors in segmentation or guideline logic. We added this to the revised manuscript as well.

---

### Author Rebuttal · Authors · 2026-01-28

**Rebuttal:**

We had attached an updated version of the manuscript. In response to the reviewers’ comments, we added focused clarifications to the text, a small number of illustrative failure examples, and a few examples demonstrating the advantages of planning and tool usage. Additional supporting details were placed in the appendix to keep the main paper streamlined.
These changes are intended to improve clarity while preserving the original scope and conclusions of the work.

**Supporting Material:**

/attachment/0a1e88222f1f4c69205a5a0e8479bc5c141951aa.pdf

---

### Meta-Review · Area_Chair_AmCn · 2026-02-03

**Recommendation:** Accept (Poster)
**Confidence:** 5

**Metareview:**

There was a consensus amongst reviewers to accept this work initially, which was further strengthened by the evidence provided in the rebuttal. However, as the authors and reviewers both agree that this work is not close to deployable as currently executed, please add a detailed discussion of limitations as a subsection in the final paper

---

### Decision · Program_Chairs · 2026-02-13

Accept (Poster)